# BTN3A: A Promising Immune Checkpoint for Cancer Prognosis and Treatment

**DOI:** 10.3390/ijms232113424

**Published:** 2022-11-03

**Authors:** Abdou-samad Kone, Saadia Ait Ssi, Souha Sahraoui, Abdallah Badou

**Affiliations:** 1Laboratory of Immuno-Genetics and Human Pathologies, Faculty of Medicine and Pharmacy of Casablanca, Hassan II University, Casablanca 20000, Morocco; 2Mohammed VI Center of Oncology, CHU Ibn Rochd, Faculty of Medicine and Pharmacy of Casablanca, Hassan II University, Casablanca 20000, Morocco

**Keywords:** BTN3A, T lymphocytes, cancer, immune system, prognostic factor

## Abstract

Butyrophilin-3A (BTN3A) subfamily members are a group of immunoglobulins present on the surface of different cell types, including innate and cancer cells. Due to their high similarity with the B7 family members, different studies have been conducted and revealed the involvement of BTN3A molecules in modulating T cell activity within the tumor microenvironment (TME). However, a great part of this research focused on γδ T cells and how BTN3A contributes to their functions. In this review, we will depict the roles and various aspects of BTN3A molecules in distinct tumor microenvironments and review how BTN3A receptors modulate diverse immune effector functions including those of CD4+ (Th1), cytotoxic CD8+ T cells, and NK cells. We will also highlight the potential of BTN3A molecules as therapeutic targets for effective immunotherapy and successful cancer control, which could represent a bright future for patient treatment.

## 1. Introduction

Therapeutic strategies for cancer have garnered a large amount of interest in recent years. Along with conventional therapies, such as radiotherapy and chemotherapy, immunotherapy seems to be a promising approach to cancer management [1]. The exhaustion of effector T cell functions via inhibitory signaling pathways triggered by certain immune checkpoints represents the most effective strategy to promote cancer cell escape and growth [2,3]. Therefore, therapies based on immune-checkpoint blockade(s) (ICB), such as ipilimumab and nivolumab, have achieved promising results and offer hope [4,5,6].

However, tumor microenvironment (TME) heterogeneity and toxicity/resistance occurrences limit the scope of this ICB [7]. Studies have revealed that the success rate does not exceed 15 to 30% in different human cancers, such as melanoma, hepatocellular carcinoma, and non-small cell lung cancer [8,9,10,11,12]. To overcome this issue, different approaches must be considered, including the implementation of personalized treatment, and the identification of additional genetic mutations, biomarkers, or new target molecules involved in tumor mechanisms and immune cell activities [13]. Thus, several studies have initiated investigations on new families of immune checkpoints in order to clarify their specific roles in TME and immune cell modulation [14,15,16].

Based on previous studies, BTN3A molecules (also termed CD277), are expressed in different types of immune cells, including T cells, natural killer cells, dendritic cells, and monocytes [17,18,19], and some cancers, such as ovarian cancer, gastric cancer, pancreatic cancer, breast cancer, and colorectal cancer [20,21,22,23,24].

Among all 13 BTNs that are known in humans, only the BTN3A subfamily of butyrophilins are expressed by tumor cells [25] as well as by all cells of the immune system, including T cells, B cells, monocytes, dendritic cells, and natural killer (NK) cells. This broad expression pattern has made this group of butyrophilins among the best-studied subtypes [19,26,27]. Another particularity of the BTN3A molecules is that the biological impacts of the various inhibitor and stimulator antibodies of BTN3 on immune responses are quite different. In addition, the molecules of the BTN3A subfamily are specific for primates and other eutherian species and are not present in rodents [28,29]. This feature of BTN3A molecules has piqued the interest of researchers because their potential biological/immunological roles are characteristic of a well-defined species. Similar to γδ T cells, BTN3A isotypes are found at the border of innate and adaptive immune responses. Highlighting their roles in modulating CD4+, CD8+, monocyte, and NK cell activity will undoubtedly lead to a better understanding of the communication between innate and adaptive immune cells and, as a result, better utilization of potential therapeutic strategies.

The literature on BTN3A primarily focuses on γδ T cells [23,28,30,31,32], which may lead to an underestimation of this molecule’s potential in modulating the immunological activities of CD4+, CD8+, monocytes, and NK cells. Therefore, it is important to provide a comprehensive overview of the BTN3A molecule’s effect on CD8 T cells, CD4 (Th1) T cells, and NK cells, summarize differential prognoses, clinical implications, therapeutic potential, and downstream signaling cascades based on the cellular contexts of BTN3A molecules in cancers.

## 2. Genetic Profile and Structural Organization

Butyrophilin genes are encoded in clusters and have been known to share the same Phylogenetic tree with other B7 family members [17]. The genetic profile of *BTNs* shows a set of genes localized in the telomeric region of chromosome 6p22, close to the major histocompatibility complex (MHC) class I genes. To date, 7 *BTNs* have been described in humans: *BTN1A1*, *BTN2A1*, *BTN2A2*, *BTN2A3*, *BTN3A1*, *BTN3A2*, *BTN3A3*, and 6 *BTNLs*: *BTNL2*, *BTNL3*, *BTNL8*, *BTNL9*, *BTNL10*, and *SKINT-like* [33,34].

It has been reported that *BTN1A1* and *BTN2A* are the only common variants found in both mice and humans. However, BTN3A molecules have generated great interest due to their specificity for humans and have been classified into three isoforms, BTN3A1, BTN3A2, and BTN3A3 [33,34]. The extracellular domains of these paralogous genes display very high degrees of similarity (>95%) [19,27]. The structural organizations of BTN3A1, BTN3A2, and BTN3A3 receptors consist mainly of two extracellular domains (IgV and IgC domains), a transmembrane domain, and the B30.2 intracellular domain. However, BTN3A2 does not share this B30.2 intracellular domain and expresses only the transmembrane and extracellular regions [35] (Figure 1).

The B30.2 domain has been shown to contain the tripartite motif (TRIM) family proteins, which could act as pattern-recognition receptors, such as toll-like receptors (TLRs) or nucleotide-binding oligomerization domain (NOD)-like receptor proteins [36,37,38,39]. It is well-known that these abovementioned structures have a high affinity for infection-associated molecules, such as pathogen-associated molecular patterns (PAMPs) or cell damage (DAMP) [40]. Therefore, evidence has highlighted the involvement of the B30.2 domain in the interaction with some endogenous (DAMP) and exogenous (PAMP) molecules. The signaling pathway ensuing from this interaction could play a key role in stimulating immune responses [28,37].

## 3. Clinical Significance of BTN3A Molecule Expression

The failures of several immunotherapeutic approaches have shifted the focus toward the identification of more relevant prognostic markers for accurate therapy and detection of very early-stage cancers. These prognostic markers could be the expressions of specific genes or transcript alterations or the level of particular proteins in body fluids. Thus, the prediction of treatment response or the evaluation of disease progression is made possible by the measurements of these markers [41].

Studies revealed that *BTN3A* expression can be detected in the spleen, heart, placenta, pancreas, lymph node, trachea, adrenalin gland, ovary, small intestine, appendix, thymus, and lymphocytes [17,25,26,27].

In some cases, BTN3A expression, especially in the plasma, could help predict the outcomes of distinct therapies. Thus, depending on the type of cancer, differential prognoses and clinical implications of BTN3A molecules are presented in Table 1.

T cells are considered the leading players in the immune response. Being able to identify the specific roles of immune checkpoint proteins on T cell activities is crucial to enhancing immune responses, preventing cancer progression, and improving patient survival [42,43]. Hence, a prominent role of BTN3A in regulating different cellular and molecular immune actors has been explored in various cancer tissues.

**Table 1 ijms-23-13424-t001:** Differential prognosis and clinical implication of BTN3A molecules.

BTN3A Isoforms	Cancer Types	Detection Methods	Clinical Significance	Prognosis	References
BTN3A1	Metastatic renal cell carcinoma (MRCC);	Baseline plasma levels of soluble BTN3A1	Predicting PD-1(Programmed cell death protein 1) immunotherapy response	Favorable	[44]
	non-small cell lung cancer (NSCLC)	mRNA/protein expression profile	Prognostic biomarker	Favorable	[45]
	Pancreatic ductal adenocarcinoma (PDAC)	Plasma levels of soluble BTN3A1	Biomarkers that reflect the progression and prognosis of PDAC	Unfavorable	[22]
	Breast cancer/non-small cell lung cancer (NSCLC)	mRNA/protein expression profile	Prognostic biomarker	Favorable	[46]
	Low-grade glioma (LGG)	GEPIA (Gene Expression Profiling Interactive Analysis) datasets		Unfavorable	[46]
	Melanoma	GEPIA datasets		Favorable	[46]
BTN3A2	Pancreatic ductal adenocarcinoma (PDAC)	Transcriptional level	Prognosis marker	Unfavorable	[22]
	Epithelial ovarian cancer (EOC)	Protein expression	Prognosis marker	Favorable	[20]
	Lung adenocarcinoma (LUAD)	mRNA/protein expression profile	Prognosis marker	Favorable	[46]
	Pancreatic ductal adenocarcinoma	Culture cell/flow cytometry	Prognosis marker	Unfavorable	[22]
	Breast cancer	Oncomine database	Prognosis marker	Favorable	[47]
	Brain cancer	Oncomine database		Unfavorable	[47]
BTN3A3	Non-small cell lung cancer (NSCLC)	mRNA/protein expression profile	Prognosis marker	Favorable	[48]
	Ovarian cancer (OC)	Protein expression Cancer	Prognosis marker	Favorable	[49]
	Gastric cancer	Therapeutics Response Portal (CTRP)/The Cancer Genome Atlas (TCGA)	Prognosis marker	Favorable	[50]

## 4. BTN3A and Immune-System

The immune system is a large and complex organization that brings together different cells, tissues, organs, proteins, and other biological components involved in the defense of the organism [51,52]. The immune cells interact via a multitude of signaling pathways, and for certain cells, the activation occurs after different co-stimulatory signals [53]. It has been shown that the T cell receptor (TCR) is involved in the first activation signal while the second signal is delivered by certain B7 family molecules [54,55,56].

Bioinformatic analyses via the TCGA (The Cancer Genome Atlas) database have revealed the involvement of the *BTN3A2* co-expression gene in many biological processes, including the activation of T cell receptor signaling pathways, cytokine receptors signaling pathways, and immune infiltration of CD8+ T cells, Th1 cells, dendritic cells (DCs) [47]. Strikingly, *BTN3A2* was positively correlated with T cell transcription factors, and anti-tumoral mediators, such as *TBX21*, *STAT1*, *IFNγ*, and *GZMB*, respectively, in triple-negative breast cancer (TNBC), compared with other breast cancer subtypes, which suggests that high expression of *BTN3A2* in TNBC patients could significantly increase the number of cytotoxic cells CTLs, Th1, and DC in the tumor microenvironment [47]. In lung adenocarcinoma (LUAD) and epithelial ovarian cancer (EOC), it has been suggested that *BTN3A2* expression was positively correlated with CD4+ T cell, neutrophil, B cell, and macrophage infiltration to the tumor area [20,45]. On the other hand, *BTN3A3* expression was positively correlated with the density of CD8+T cells, anti-tumor immune response, and less invasive phenotype in non-small cell lung cancer (NSCLC) patients [48]. The TME is driven by a panoply of inflammatory signals [57]. Research shows that *BTN3A* expression is increased by pro-inflammatory cytokines, such as IFNγ and TNFα [26,47,58].

The mechanism underlying the involvement of BTN3A molecules in immune cell infiltration and CD4+/CD8+ T cell activities is not yet well understood. However, it might be related to the specificity of the ligand/receptor binding domains expressed on antigen presenting cells (APCs), and T cells, respectively, and the type of signal transmitted as a result of this binding [20]. BTN3A receptors expressed on APCs could promote negative co-stimulatory functions following the binding on the ligand expressed by T cells. As a matter of fact, BTN3A1 expressed on APCs has been shown to restrain human T cell proliferation in ovarian carcinoma microenvironments [19,20,25]. On the other hand, a positive co-stimulatory function could be induced when BTN3A1 is expressed on T cells or epithelial cells as a receptor [20,25].

### 4.1. BTN3A Co-Inhibitory Function

Interaction with different BTN3A receptors induces distinct effects on T cell functions, depending on both heterogeneity and the binding domain. Yamashiro et al. have generated the 232-5 mAb specific to the variable site of the extracellular region of BTN3A molecules. This variable region includes amino acids 35–139 from the N-terminal position and displays a high affinity to 232-5 mAb.

Significant inhibition of CD4+/CD8+ T cell proliferation and IL-4 and IFNγ production has been reported after treatment of Human peripheral blood mononuclear cells (PBMCs) with BTN3A3-specific inhibitory monoclonal antibody 232.5 [26], suggesting the inhibition property of 232-5 mAb. Interestingly, the same inhibitory effect has been observed on CD25+ Treg cell-depleted PBMCs, suggesting that this suppression is independent of CD25+ Treg cells and involves both 232.5 mAb and the variable region of the extracellular region of the BTN3A molecule [26]. Furthermore, significant expression of BTN3A on MHC-II+ ovarian cancer-associated DC/macrophages was found to be associated with both inhibitions of the anti-tumor T cell response and production of proinflammatory cytokines, such as IL-2, IFNγ, and TNFα [25]. However, elevated BTN3A expression was associated with an increase in IL-17 and IL-6 production [25]. Cellular FLICE-like inhibitory protein (c-FLIP) through interaction with pro-caspase 8, represents one of the key components that could explain the altered function of T cells. The Mitogen activated protein kinase (MAPK)/Extracellular-signal-regulated kinase (ERK) and Phosphatidyl-inositol 3 kinase (PI3K)/serine/threonine protein kinase (AKT) cascade represent two major signaling pathways that regulate the transcription factor Nuclear factor kappa B (NF-κB) [59,60,61,62]. It has been shown that this anti-apoptotic mediator c-FLIP promotes T cell maturation and survival via the activation cascade ERK/NF-κB [63,64,65]. It would seem that BTN3A suppresses c-FLIP expression and, thus, contributes to the inhibition of T cell activities [25]. The negative co-stimulatory function of BTN3A has also been highlighted in a recent study using engineered antigen-presenting cells expressing BTN3A1 (BTN-K32 aAPCs) [66]. CD4+/CD8+ T cell proliferation, activation, and IFNγ production were suppressed after 6 days of co-culture with HLA-A2+ BTN3A1-K32 cells. Furthermore, CD4+/CD8+ T cells have recovered their function after treatment with the CTX-2026 mAb that binds to the IgV extracellular domain of BTN3A1 [66]. The involvement of the transmembrane PTPase CD45 could be behind this inhibitory effect of BTN3A1. CD45 is well recognized to play a key role in T cell activation through the TCR signaling pathway [67,68]. BTN3A1 has been shown to bind the N-mannosylated residues of CD45 and, thus, could trigger co-inhibitory TCR signaling. Interestingly, CRISPR-mediated deletion of CD45 in CD4+/CD8+ T cells has completely restored their primary function and obviates CTX-2026 mAb utilization [66].

BTN3A co-inhibitory function encompasses a multitude of structural and biological parameters, including TME complexity, binding domain specificity, location, and different signaling pathways. This suggests the key role of BTN3A molecules in modulating the immune system, which requires further research to better understand the different facets of these receptors.

### 4.2. BTN3A Co-Stimulatory Function

The involvement of BTN3A molecules in positive co-stimulation of TCR signaling has been investigated by using a newly generated mAb clone, 20.1 [27]. Messal et al. have demonstrated the potential of BTN3A (CD277) molecules to induce a significant increase in IL-2 and IFNγ production by CD4+ T cells after different cell culture conditions. CD4+ T cells were stimulated with CD3 plus CD28 mAbs or CD3 plus CD277 mAbs or CD3 mAb plus IgG1 (control conditions) and, surprisingly, IFNγ production was significant in CD3 plus CD277 stimulation compared to CD3 plus CD28 co-activation [19]. Thus, it would appear that BTN3A receptor stimulation by 20.1 mAb results in T cell proliferation and cytokine production in a dose-dependent manner [19]. PI3K/AKT and MAPK/ERK signaling pathways appear to be involved in the co-stimulatory property of BTN3A1. These pathways regulate cell growth, proliferation, survival, and invasion after phosphorylation cascades [69,70]. Stimulation by CD3 plus CD277 mAbs may have resulted in the phosphorylation of AKT and ERK and subsequent positive stimulation of CD4+ T cells [19]. BTN3A1 and BTN3A2 were stimulated on the KGHYG-1 NK cell line ‘nucleofected’ with constructs encoding for flag epitope which tagged BTN3A1 and BTN3A2. The construction of the flag epitopes was performed by deleting the signal peptide sequences from WT full-length human cDNA of BTN3A1 and BTN3A2. The results suggest that BTN3A1 stimulation increases IFNγ production, whereas, BTN3A2 stimulation decreases the NKp30-induced IFNγ production [19]. These results could be explained by the NK cell that expresses mainly BTN3A2, which lack the B30.2 intracellular domain. BTN3A2 may be considered a putative receptor devoid of co-signaling function in NK cells compared to well-known co-stimulatory (DNAM-1) and co-inhibitory (NKG2A) molecules [19]. In addition, it has been demonstrated that transfection of NSCLC cell line with siRNA to knock-down *BTN3A3* as well as patients with low BTN3A3 expression displayed invasion, migration, and proliferation of NSCLC cells [48]. This study underlines the crucial role of BTN3A3 since patients with high BTN3A3 expression have shown increased CD4+/CD8+ T cell infiltration and better clinical outcome [48]. Recently, another monoclonal antibody, ICT01 has been developed with a similar affinity for the three BTN3A isoforms. De Gassart et al. have reported that BTN3A+ Vγ9Vδ2 T cell activation by ICT01 induced apoptosis of multiple tumor cell lines and primary tumor cells without affecting normal cell viability. It has been reported in melanoma patients that ICT01 may promote immune cell infiltration within the tumor microenvironment. Moreover, preliminary results of phase 1/2a performed on patients with various types of advanced stages of solid tumors showed that ICT01 was well endured and pharmacodynamically active. In addition, a co-culture of PBMCs with PC3 or HT29 cell lines promoted Vγ9Vδ2 T cell expansion and cancer cell mortality [71].

Therefore, the CD277 co-stimulatory pathway may differentially contribute to the regulation of various immune cell response (see Table 2). Thus, the co-stimulatory function of BTN3A could turn out to be a determinant in the therapeutic strategies. It is well established that the percentage of CD8+ T cell infiltration is crucial for the prediction and success of certain therapies including immunotherapy [72,73,74,75,76]. Thus, BTN3A receptors through their involvement in CD4+/CD8+ T cell activation and high pro-inflammatory cytokine production, are considered the main actors in immune cell infiltration and cancer treatment.

Based on previous studies, the accurate functions of BTN3A molecules in immuno-modulation are quite disparate. A thorough understanding of the different BTN3A signaling pathways through their putative binding partners is necessary to shed light on BTN3A functionality in the immune system.

## 5. BTN3A and Putative Ligand

To date, several studies have attempted to identify the exact counter-receptor of BTN3A molecules. BTN3 ligand has been found in leukemia and solid tumor cell lines, such as HeLa and MCF-7, Raji, C91, HUT78, and JA16 [27,77]. Using the CD277-Fc fusion protein on C91 cells, a clone population termed C91T3.3 was generated. The binding stability of the CD277-Fc fusion protein with the potential ligand was confirmed by flow cytometry. In addition, the co-immunoprecipitation assay after cross-linking confirmed the presence of the putative ligand of BTN3A molecules. However, the characterization of the exact ligand for BTN3 could not be achieved due to the possible loss of the CD277 counter-receptor heterodimerization during extraction of the corresponding bands from the PAGE-SDS gel for mass spectrometry detection [77]. Detection strategies could not be appropriate and should be optimized for the correct characterization of this counter-receptor. Research is still ongoing concerning the accurate detection of BTN3A ligands [77]. In the same vein, Compte et al. excluded PD-L1, CTLA-4, CD28, ICOS, and BTLA as candidate binding partners of BTN3A on T cells and demonstrated that the IgG-V domain of BTN3A is mainly involved in the interaction with the counter-receptor [27].

Interestingly, more recent studies have shed light on the potential new receptors for BTN3A1 and BTN3A3. It has been shown that the LSECtin protein acts as a co-inhibitory ligand through the interaction with BTN3A1. Indeed, through the immunological ELISA test, interactions between LSECtin, and BTN3A1 have been highlighted. This led to an impairment of T cell activation and proliferation, as well as a decrease in the production of pro-inflammatory cytokines, such as IFN-γ, IL-2, IL-17, and TNF-α. Moreover, anti-BTN3A1 antibody administration has partially restored T cell activity. Therefore, the LSECtin/BTN3A1 axis appears to be a promising therapeutic target [78]. BTN3A3 has been also found to interact with LSECtin on tumor-associated macrophages and contribute to the promotion and survival of breast cancer cells [79].

This interaction was confirmed using HEK293 cells transfected with a commercial human cDNA library and devoid of known LSECtin receptors. cDNA library-expressing HEK293 cells were generated after transfection and BTN3A3 was recognized as the LSECtin-binding receptor after screening of LSECtin-binding cells plasmids. It would seem that the extracellular region (IgC/IgV) of BTN3A3 is imperative for LSECtin interaction while the intracellular domain is involved in the initiation of the The Janus kinase ( JAK2)/ Signal Transducer And Activator Of Transcription3 (STAT3) signaling pathway and phosphorylation cascades [79]. Research shows that JAK/STAT3 signaling is strongly involved in cancer stem cell (CSC) promotion, epithelial–mesenchymal transition (EMT), and breast cancer cell proliferation [80,81]. Therefore, the interaction between BTN3A3 and LSECtin contributes to breast cancer progression by activating the JAK-STAT pathway [79].

Given the structural organization of BTN3A, the intracellular domain represents a potential binding site, particularly for BTN3A1 and BTN3A3. Indeed, this domain distinguishes the three isoforms and is involved in the integrity of cell cytoskeleton via the interaction with the plakin family proteins such as periplakin. However, the di-leucine motif is only found in the cytoplasmic tail of BTN3A1, which is crucial for interaction with the plakin family members. Therefore, BTN3A2 or BTN3A3 do not share this functionality ensuing from this binding [23,82]. It has been shown that phosphoantigens, such as IPP (isopentenyl-pyrophosphate), DMAPP (dimethylallyl-pyrophosphate), or HMB-PP((E)-4-hydroxy-3-methyl-but-2-enyl pyrophosphate), as well as biochemical intermediates of isoprenoid biosynthesis, promote the co-stimulatory activity of BTN3A through the B30.2 intracellular domain [28,37].

## 6. BTN3A Signaling Pathways

T cell activation by butyrophilins 3 may occur through the antigen-presenting model, the release of phospho-antigen to the extracellular region, and the formation of the BTN3A1-phospho-antigen complex that binds to γδ TCR activation [32]. This could also involve an inside-out signaling model, through which the phospho-antigen attaches to the N-terminal part of the intracellular B30.2 domain of BTN3A1 with high affinity to a positively charged pocket to elicit a (β-α) conformational transition of H351 residue, then the BTN3A1-phospho-antigen complex displaces to juxtamembrane region. This complex immobilization increases the attraction between the extracellular domain of BTN3A1 and the γδ TCR which induces γδ T cell activation [36,83,84].

It has been recently shown that 350 and 391 are the two crucial residues responsible for H351 transition in the B30.2 domain, and the mutation of w391 decreases phospho-Antigen-B30.2 domain binding. Further, the residues of positively charged pocket are highly conserved in the BTN3A1 B30.2 domain compared to other proteins [83,84]. The requirement of BTN3A2 and BTN3A3 in T cell activation was also explored. Results indicated that these two isotypes optimize the effect of BTN3A1 by controlling the suitable routing, dynamic, and stability of BTN3A1 [85].

Recognizing cell expression of the phosphorpho-antigen-BTN3A1-B30.2 domain complex by T cells induces an immunological synapse [86] establishment, which leads to signal transduction and activation of different signaling pathways. Anti-CD277mAb provokes the phosphorylation of phosphoinositide-specific phospholipase (PLCγ2) in TCR Vγ9γδ cells, and the activation of AKT by the phosphorylation of its residues by MTORC2 and PDR1. Thereby, modulation of metabolism and cell survival upon NF-кB activation. On the other hand, TCR activation elicits equally the MAPK pathway by increasing intracellular T cell phosphorylation of ERK1 and P38 after anti-CD277mAb treatment [87] (Figure 2).

## 7. Perspectives and Conclusions

In light of the preceding, we can state that BTN3A molecules play a significant role in modulating CD4+, CD8+, monocytes, and NK cell activity. Their roles as biomarkers in different types of cancer make them critical players in facilitating the prognosis of patients.

However, many questions remain unanswered about BTN3A receptors, particularly their specific ligands that would induce a co-stimulatory/inhibitory effect on T lymphocyte activities.

Do T cell immunomodulatory effects (upon the interaction of BTN3A with potential ligands or different antibodies) occur under specific physiological conditions? Which IgV extracellular-binding domains are involved in this interaction? What is the distinguishing feature of each antibody?

BTN3A receptors are classified into two major domains: the extracellular domain and the B30.2 domain (primarily for BTN3A1/3). So, which part of the BTN3A receptor (blocking or enhancing a co-stimulatory function) would be more advantageous to target? 

What role does the B30.2 domain play in the signaling pathways and cascades triggered by the putative ligand’s interaction with the extracellular region of BTN3A receptors on T cells? How do agonistic or antagonistic BTN3A-targeting molecules affect adaptive immunity in vivo, and is combination immunotherapy with checkpoint blockade therapies, such as anti-PD-1 blocking antibodies and BTN3A agonists or antagonists, beneficial?

In summary, several pieces of evidence pinpoint the crucial role of BTN3A molecules on T cell functions within the tumor microenvironment in solid tumors. Current research on the signaling pathways by which BTN3A expression affects CD4+, CD8+, monocytes, and NK cell functions, is limited. Further studies should be pursued to shed light on the exact activating and/or inhibiting roles of BTN3A molecules. Answering these critical questions would elucidate how BTN3A proteins could be used in immunotherapy to improve the anti-tumor immune response.

## Figures and Tables

**Figure 1 ijms-23-13424-f001:**
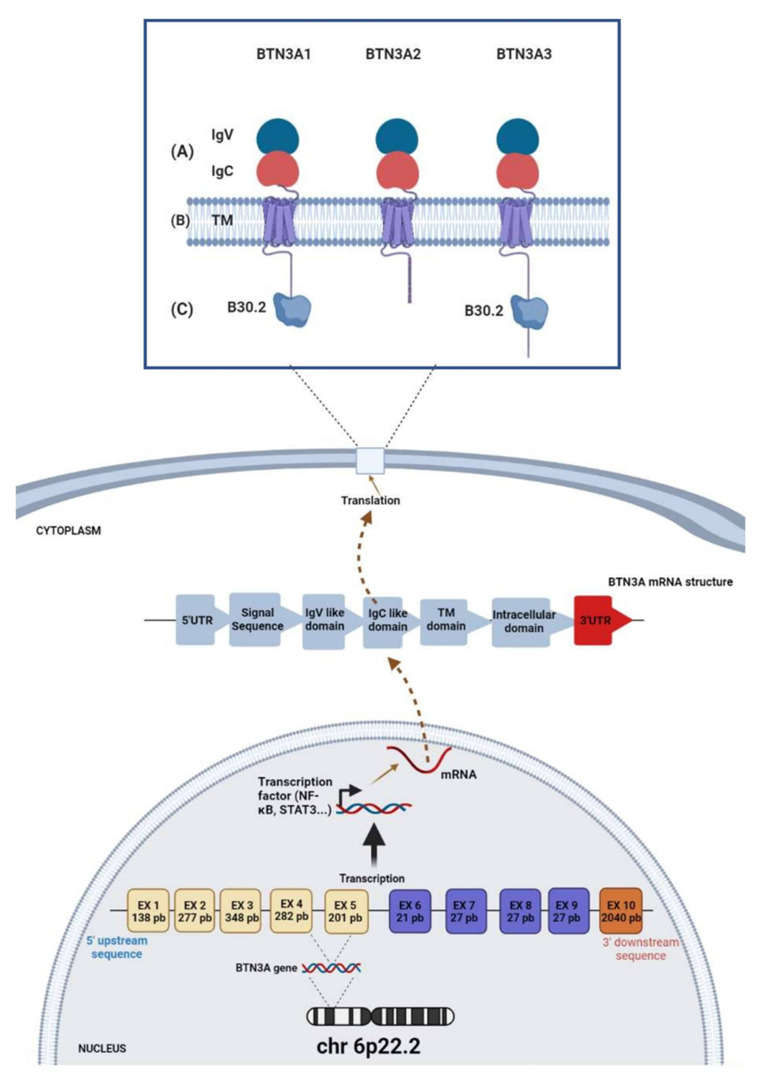
Schematic representation of the *CD277/BTN3A* gene, mRNA, and protein structural domains. (**A**) The extracellular domain of interaction with the ligand. (**B**) The transmembrane domain. (**C**) The intracellular domain. BTN3A1 and BTN3A3 share the B30.2 domain involved in the interaction with the phosphoantigens.

**Figure 2 ijms-23-13424-f002:**
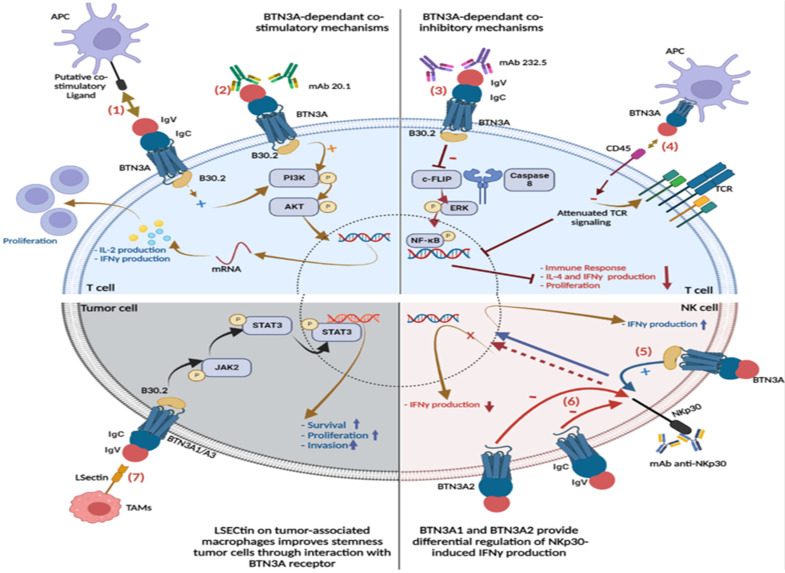
BTN3A interaction patterns and signaling pathways. BTN3A-dependent co-stimulatory mechanisms: (1) engagement of BTN3A with its putative ligand on APC and/or (2) treatment of BTN3A with CD277 mAb 20.1 provides a co-stimulatory T cell activation signal that induces production of IL-2 and IFNγ as well as T cell proliferation via the PI3K/AKT signaling pathway. BTN3A-dependent co-inhibitory mechanisms; (3) interaction of BTN3A with a specific mAb 232.5 suppresses c-FLIP expression. The cellular inhibitory protein type FLICE-like (c-FLIP) is well-known to promote the maturation and survival of T cells through interaction with pro-caspase, which activates the activation cascade ERK/NF-κB. The inhibition of c-FLIP expression by BTN3A triggers an inhibiting signal that interferes with the production of IFNγ, IL-4, and the proliferation of T cells. (4) BTN3A1 binds the N-mannosylated residues of CD45 and, thus, elicits the inhibition of the TCR signal on T cells. CD45 is well recognized to play a key role in T cell activation through the TCR signaling pathway. The BTN3A1 and BTN3A2 provide differential regulation of NKp30-induced IFNγ production: (5) The co-engagement of BTN3A1 with NKp30 after treatment with anti-NKp30 mAb modulates the NKp30-induced IFN-γ production, (6) whereas the lack of B30.2 domain in the BTN3A2 structure impair the co-engagement of BTN3A2 with NKp30 and thus lead to the attenuation of the NKp30-induced IFN-γ production. LSECtin on tumor-associated macrophages improves the stemness of tumor cells through interaction with BTN3A receptor: (7) BTN3A1/2 Interaction with LSECtin in tumor cells actives the JAK2/STAT3 signaling pathway, which promotes cancer cell survival, invasion, and proliferation.

**Table 2 ijms-23-13424-t002:** Therapeutics potential of BTN3A molecules in cancers.

Cell Subsets Expressing BTN3A	Detection Methods	Impact on Effector Activities	Therapeutics Potential	References
CD8 T cell	PBMCs/culture cell/flow cytometry	Attenuation of CD8 T cell proliferation and IFN production	Blocking of BTN3 signal transduction or destruction of BTN3 mRNA with small interfering RNA may be applicable for patients with tumors	[26]
CD4 T cell		Attenuation of CD4T cell proliferation and IL-4 production		
Regulatory T cell		Attenuation of regulatory T cell proliferation		
CD4 T cell	PBMCs and lymph nodes/culture cell/flow cytometry	T-cell activation	Positive immunomodulators of T cell responses, which may ensure good responses to immunotherapies	[19]
NK cell		Decreased IFN_γ_ production by NK cell upon specific engagement of BTN3A2		

## Data Availability

Not applicable.

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
