# Peer review of "BTN3A: A Promising Immune Checkpoint for Cancer Prognosis and Treatment"

_ijms, 2022, doi:10.3390/ijms232113424_

Round 1
Reviewer 1 Report
The review article entitled "BTN3A: a promising immune checkpoint for cancer prognosis and treatment" is very interesting. However, there is a similar article already published online.
Sihan Chen, Zhangyun Li, Wenyi Huang, Yanyan Wang, Shaohua Fan. Prognostic and Therapeutic Significance of BTN3A Proteins in Tumors. Journal of Cancer 2021; 12(15): 4505-4512.
Therefore, authors should mention the novelty of their review article with clear explanation.
Author Response
Response to Reviewer 1 Comments
Point 1: The review article entitled "BTN3A: a promising immune checkpoint for cancer prognosis and treatment" is very interesting. However, there is a similar article already published online.
Sihan Chen, Zhangyun Li, Wenyi Huang, Yanyan Wang, Shaohua Fan. Prognostic and Therapeutic Significance of BTN3A Proteins in Tumors. Journal of Cancer 2021; 12(15): 4505-4512.
Therefore, authors should mention the novelty of their review article with clear explanation.
Response 1: Thank you for raising this comment. The review article entitled “Prognostic and Therapeutic Significance of BTN3A Proteins in Tumors” has focused on how BTN3A contributes to the activation of γδ T cells. This information is correct, but the fact that the entire literature on BTN3A1 is mainly focused on γδ T cells may underestimate the potential of this molecule in modulating the immunological activity of CD4+, CD8+, Monocytes, and NK cells. This part was one of our purpose, so we highlighted it in our manuscript. Moreover, to our knowledge, our paper is the first to provide details of the potential signaling and molecular pathways triggered by the interaction of BTN3A with its putative ligand in different immune cells other than γδ T cells.
Reviewer 2 Report
Reviewer’s report
MS: ijms-1898136
The review article wrote by Kone et al., delivers a timely and useful information to support BTN3A as a promising therapeutic target for cancer immunotherapy. This review could be an important contribution to the field. However, as in its current form, it cannot be considered for publication in the IJMS. I have several comments which I believe would help improving the quality of this review.
Comments
1. I may be wrong, but I feel the introduction and the section 2 did not provide sufficient information to support a rationale of this review: why BTN3A, but not other BTNs, should be focused on. Perhaps, a table summarizing all 13 BTNs and their tissue distribution, functional roles and/or disease associations could address this issue.
2. Lines 112-118: I understand the authors cited a result in Cai, et al, (Journal of Cellular Biochemistry. 2020, 121, 2643–2654). But it would be nice if this review provides a figure to show BTN3A expression across multiple cancer types. This new figure would help the reader understand the potential importance of BTN3A as a biomarker and/or therapeutic target in cancers.
3. The section 3 (Clinical significance of BTN3A molecule expression) has only one subsection 3.1 (BTN3A as a biomarker). I believe this section should describe BTN3A as a therapeutic target as the subsection 3.2 as well. This subsection would also serve as a nice transition for the sections 4-6.
4. Figure 2 is very informative and eye-catching. But there is an issue on the figure legend system. The legend of (1) – (7) is very useful, but that of A – D did not contribute much, and sometime causing a confusion. Replacing the legend of A – D with the mode of BTN functions (e.g., BTN3A-dependent co-stimulatory mechanism; BTN3A-dependent co-inhibitory mechanism, etc) can improve the quality of this important figure.
5. This reviewer humbly suggests the authors provide their perspectives on the future direction of BTN3A research.
Author Response
Point 1: I may be wrong, but I feel the introduction and the section 2 did not provide sufficient information to support a rationale of this review: why BTN3A, but not other BTNs, should be focused on. Perhaps, a table summarizing all 13 BTNs and their tissue distribution, functional roles and/or disease associations could address this issue.
Response 1: Thank you for this relevant remark, indeed, it would be a good idea to add a summarizing table but different reviews have provided this kind of tables “Lucie Abeler et al, Butyrophilins: an emerging family of immune regulators, Trend in immunology, 2012”. However, we would like to draw the scientific committee's and the reader's attention to an important issue.
- To begin with, most publications have investigated the role of BTNs in the activation of V9V2 T cells under the guise that they are the only subpopulations that are activated by posphoantigens, but there is no paper that denies this in relation to other lymphocyte populations.
- BTN3A is only found in human species and has no murine counterpart, so most research has focused on human tissues or cell lines. As a result, we would like to investigate the role of different BTN3 isoforms in the activation and intracellular signaling of CD8, CD4, and NK cells.
- Given that immunotherapy success is heavily reliant on the lymphocytes infiltration,
particularly TCD8, our work would allow reconsideration of research directions which is based primarily on V9V2 T cells.
In the manuscript, we have added a new paragraph where we highlighted the rational of our review (see introduction section).
Point 2: Lines 112-118: I understand the authors cited a result in Cai, et al, (Journal of Cellular Biochemistry. 2020, 121, 2643–2654). But it would be nice if this review provides a figure to show BTN3A expression across multiple cancer types. This new figure would help the reader understand the potential importance of BTN3A as a biomarker and/or therapeutic target in cancers.
Response 2: We agree with this remark, we included now a table that explains the prognosis and clinical implications of the various BTN3 isoforms in patients with various cancers in the new version of the manuscript (see table 2).
Point 3: The section 3 (Clinical significance of BTN3A molecule expression) has only one subsection 3.1 (BTN3A as a biomarker). I believe this section should describe BTN3A as a therapeutic target as the subsection 3.2 as well. This subsection would also serve as a nice transition for the sections 4-6.
Response 3: Thank you for this remark, indeed, we deleted now the subsection title named “3.1 BTN3A as a biomarker” and we conserved the main title “3. Clinical significance of BTN3A molecule expression”
Point 4: Figure 2 is very informative and eye-catching. But there is an issue on the figure legend system. The legend of (1) – (7) is very useful, but that of A – D did not contribute much, and sometime causing a confusion. Replacing the legend of A – D with the mode of BTN functions (e.g., BTN3A-dependent co-stimulatory mechanism; BTN3A-dependent co-inhibitory mechanism, etc) can improve the quality of this important figure.
Response 4: Thank you for this relevant remark, the legend of (A-D) was all changed by the BTN3A functions in the new version of the manuscript.
Point 5: This reviewer humbly suggests the authors provide their perspectives on the future direction of BTN3A research.
Response 5: We agree with this remark, All the questions we raised while writing this review have been incorporated into the new version of the manuscript (see section 8).
Reviewer 3 Report
The draft article "BTN3A: a promising immune checkpoint for cancer prognosis and treatment" reviews the recent updates regarding the immune cell signalling within the tumour microenvironment (TME).
At first glance, the authors wanted to focus on T cells and the molecule-cell interactions. However, the article mainly focuses on signalling and molecular pathways with some relevance to CD4/CD8 cells. Here are some comments to be considered and may be applied:
1- Technically, this is a review article and both authors and the journal wants to attract the attention of basic and clinical scientists. I would recommend reducing the text and adding figures and tables to cover the main ideas of the topic.
2- In the abstract, please clarify the paper's main focus. Using the word "conventional" may be misleading. There is a variety of CD4 and CD8 T cells. One may argue that "conventionals" were introduced first by the expression of their surface markers. We know that there are several sub-groups for T cells. If the main focus is on interaction, please mention what kind of interaction you are scrutinising.
3- As you already named some cell types, I recommend gathering all T cell types in one table and briefly mentioning their relationship with BTN3A. For example, Tregs, Tfh cells, TFR cells, Tc1, Th17, etc. You may only refer to those with at least one citation. I understand there might not be any paper for Tfh, but do your best to find some data from expression databases. Also, as you are interested in referring to therapeutic implications, draw a table and briefly elaborate your thoughts on the therapeutic aspects. Adding this sort of table is helpful for educational purposes.
4- In the first figure, also add the gene and protein structure so the reader can follow the gene-to-function pathway.
5- The quality of images is low; improve the quality if it's your res[onsibilty. Usually, "tif" files have better resolution.
Good luck
Author Response
Point 1: Technically, this is a review article and both authors and the journal wants to attract the attention of basic and clinical scientists. I would recommend reducing the text and adding figures and tables to cover the main ideas of the topic.
Response 1: Thank you for raising this point. we deleted now the subsection title named “3.1 BTN3A as a biomarker” and we conserved the main title “3. Clinical significance of BTN3A molecule expression”. We have also reduced the text, added two tables, and changed the first figure in the new version of the manuscript.
Point 2: In the abstract, please clarify the paper's main focus. Using the word "conventional" may be misleading. There is a variety of CD4 and CD8 T cells. One may argue that "conventionals" were introduced first by the expression of their surface markers. We know that there are several sub-groups for T cells. If the main focus is on interaction, please mention what kind of interaction you are scrutinising.
Response 2: We totally agree with this remark, Indeed, we aim to highlight the immunomodulatory effect of BTN3A on T cells except γδ T cells. As BTN3A is a little-studied molecule, so the first type of interaction that we have prioritized is that of BTN3A, which is expressed as a receptor on Cytotoxic CD8 + T cells (CTL), CD4 + (Th1), and NK cells and binds to its putative ligand on tumor cells or APCs or interacts with some mAbs. We also mentioned that this interaction could be accomplished by considering BTN3A as an APC receptor and its putative ligand as a T cell. Now we changed the word “"conventional" by specific immune cell subtypes in the new version of the manuscript.
Point 3: As you already named some cell types, I recommend gathering all T cell types in one table and briefly mentioning their relationship with BTN3A. For example, Tregs, Tfh cells, TFR cells, Tc1, Th17, etc. You may only refer to those with at least one citation. I understand there might not be any paper for Tfh, but do your best to find some data from expression databases. Also, as you are interested in referring to therapeutic implications, draw a table and briefly elaborate your thoughts on the therapeutic aspects. Adding this sort of table is helpful for educational purposes.
Response 3: We agree with this remark. Except that, we haven't found enough studies to compile a table that summarizes all of the subpopulations you've mentioned. Nonetheless, we created a table where we explained the relationship and differential therapeutic implications of BTN3A with major lymphocyte subpopulations (see table 1). We've also included a second table that explains the prognosis and clinical implications of the various BTN3 isoforms in patients with various cancers (see table 2).
Point 4: In the first figure, also add the gene and protein structure so the reader can follow the gene-to-function pathway.
Response 4: Thank you for raising this point, the first figure has now been changed in the new version of the manuscript (see figure 1).
Point 5: The quality of images is low; improve the quality if it's your res[onsibilty. Usually, "tif" files have better resolution.
Response 5: All figures have now been converted to “tiff” files in the new version of the manuscript.
Round 2
Reviewer 1 Report
The revised version of the manuscript entitled "BTN3A: a promising immune checkpoint for cancer prognosis and treatment" is very interesting and looks good. Authors have answered all the comments by the reviewers and justified their manuscript. Therefore, I strongly suggest to accept the manuscript without any further revision.
Author Response
Response to Reviewer 1 Comments
Point 1: The revised version of the manuscript entitled "BTN3A: a promising immune checkpoint for cancer prognosis and treatment" is very interesting and looks good. Authors have answered all the comments by the reviewers and justified their manuscript. Therefore, I strongly suggest to accept the manuscript without any further revision.
Response 1: Thank you very much for all of your comments and suggestions, which helped us to improve the quality of our work.

Reviewer 2 Report
The authors have appropriately responded to all my comments. My last suggestion is related to the format and writing style for the new section 'Perspectives'.
1. It is interesting to read a number of open questions following the conclusion of the review, which is probably making sense in some cases, but in my understanding the general format of the review article will put the perspective section before the conclusion.
2. Please also reconsider the writing style for the perspective section. In its current form this section is probably too short, covering only questions but not the answers. Perhaps, adding an opening sentence to introduce the readers that there are many questions remained to be addressed regarding BTN3A as a cancer therapeutic targets, and having a closing statement that future studies should be pursued to address them, may resolve this issue.
Author Response
Response to Reviewer 2 Comments
Point 1: It is interesting to read a number of open questions following the conclusion of the review, which is probably making sense in some cases, but in my understanding the general format of the review article will put the perspective section before the conclusion.
Response 1: Thank you for your input. To make the perspectives more consistent with the review format, we changed the style of presentation and included it with the conclusion (see section 7).
Point 2: Please also reconsider the writing style for the perspective section. In its current form this section is probably too short, covering only questions but not the answers. Perhaps, adding an opening sentence to introduce the readers that there are many questions remained to be addressed regarding BTN3A as a cancer therapeutic targets, and having a closing statement that future studies should be pursued to address them, may resolve this issue.
Response 2: Thank you for your suggestion. We have now changed the presentation style of the perspectives and included it with the conclusion to make them more consistent with the review format (see section 7).
